# An unexpected role of CLASP1 in radiation response and S-phase regulation of head and neck cancer cells

Reinout H. de Roest[1,2*], Marijke Buijze[1,2], Myrthe Veth[1,2], Klaas de Lint[2,3], Govind Pai[2,3], Martin A. Rooimans[2,3], Rob M.F. Wolthuis[2,3], Arjen Brink[1,2], Jos B. Poell[1,2], Ruud H. Brakenhoff[1,2*]

1 Amsterdam UMC location VUmc Amsterdam, Otolaryngology/Head and Neck Surgery, Head and Neck Cancer Biology and Immunology laboratory, Amsterdam, The Netherlands, 2 Cancer Center Amsterdam, Cancer Biology and Immunology, Amsterdam, The Netherlands, 3 Amsterdam UMC location VUmc Amsterdam, Human Genetics, Oncogenetics laboratory, Amsterdam, The Netherlands

* r.deroest@amsterdammumc.nl (RHR), rh.brakenhoff@amsterdammumc.nl (RHB)

## Abstract

Radiotherapy is a mainstay of treatment for head and neck squamous cell carcinoma (HNSCC), either definitive or adjuvant to surgery. Biological factors known to affect radiation response are hypoxia and DNA repair capacity, but several lines of evidence indicate that other genes and pathways in the tumor cells might be involved that have not been elucidated. Here, we report the results of a genome-wide CRISPR-Cas9 functional genomics screen in HNSCC cells to identify radiosensitizing genes. Remarkably, microtubule organizing genes were identified with *CLASP1* as most unexpected radiosensitizing hit. Clonogenic assay confirmed the radiosensitizing effect of *CLASP1* knockout. Functional analysis showed that *CLASP1* knockout has major impact during S-phase, and resulted in mitotic cells with broken chromosomes and cell death. *CLASP1* and possibly the microtubule machinery in broader sense seem involved in protection of HNSCC cells against radiation–induced DNA damage. This newly identified mechanism provides an outlook for novel treatment approaches in HNSCC.

## Introduction

Head and neck squamous cell carcinomas (HNSCCs) arise predominantly (~90%) in the mucosal lining of the upper aerodigestive tract. Worldwide around 890,000 patients are diagnosed annually with HNSCC, with a mortality of around 500,000 every year [1]. The most common risk factors for HNSCC are smoking and excessive alcohol consumption. In addition, rare genetic predisposition syndromes such as Fanconi anemia are associated with a high risk for HNSCC [2]. Last decades, human papillomavirus (HPV) infection has emerged as an etiological factor in the development of especially oropharyngeal squamous cell carcinoma (OPSCC) [3,4].

**Data availability statement:** All relevant data are within the manuscript and its Supporting information files.

**Funding:** RHB KWF-A6C7072 (Design project) KWF Dutch Cancer Society https://www.kwf.nl/en/dutchcancersociety No

**Competing interests:** The authors have declared that no competing interests exist.

HPV-positive OPSCC is nowadays considered as a distinct disease entity with unique clinical and molecular characteristics and a more favorable treatment outcome, and these tumors have been separately staged in the 8th edition of the TNM-system [5,6]. A high attributable factor for HPV is only reported for oropharyngeal cancer [7], while in oral cancer it is as low as 2–3% [8].

Radiotherapy is a cornerstone in the clinical management of head and neck cancer, in the vast majority radiotherapy is delivered as photons, but in selected cases when normal tissue toxicity is considered to be too extensive, it can also be delivered as protons [9,10]. Early stage tumors outside the oral cavity are treated with definitive radiotherapy. Definitive cisplatin-based chemoradiotherapy (CRT) has become the treatment of choice in advanced stage oropharyngeal, hypopharyngeal and laryngeal tumors, resulting in organ preservation without consequences for overall survival and locoregional control [11–13]. Surgery is still the mainstay of treatment for oral cavity tumors, but for advanced stage oral cancers, surgery is combined with post-operative radiotherapy or CRT.

The preferred CRT treatment regimen consists of high-dose cisplatin (3 times 100 mg/m$^2$) administered at baseline and every three weeks concurrent with radiotherapy [14]. Radiotherapy remains the cornerstone of this treatment regimen, while cisplatin is considered to function as radiosensitizing agent, increasing 5-year overall survival with around 7% [12,13]. Therapeutic efficacy of cisplatin is unfortunately hampered by high rates of dose-limiting toxicity such as nephrotoxicity, neurotoxicity and ototoxicity. These toxic side effects frequently lead to hospitalizations despite careful protective management, with reduction of cisplatin dose and increased risk for local failure as a consequence and long-term adverse effects [15–20]. Furthermore, patients older than 70, or patients with pre-existent renal/otological/neurological dysfunction or other severe comorbidities, are not eligible for high dose cisplatin, and these patients receive alternative but less effective treatment regimens. Therefore, new radiosensitizing agents with limited adverse events and preferably broad applicability are awaited.

The cytotoxic effect of radiotherapy relates to several mechanisms of which the most important are direct and indirect DNA damage resulting in single- and double-strand DNA breaks (SSBs or DSBs) as well as base damage [21,22]. DSBs form a small part of direct radiation-induced damage, but SSBs change in DSBs during DNA replication when left unrepaired. DSBs have the highest impact on the survival of the cell. Repair of DSBs often fails, triggering apoptotic cell death or mitotic catastrophe. DSBs could also result in premature senescence, accumulation of mutations and genomic instability [21,22]. Consequently, certain biological characteristics of the tumor determine radiation response such as the ability to repair DNA damage, to recover from radiation and repopulate between fractions, to redistribute cells in less radiation-sensitive phases of the cell cycle and many other. Besides intrinsic biological properties of the tumor cells, also hypoxia of the tumor impacts radiation response [23,24] as radiation induces formation of reactive oxygen species that also cause DNA and tissue damage.

Molecular analyses have revealed the importance of several specific genetic aberrations in the tumor cell that relate to the intrinsic sensitivity for radiation; mainly

mutations in genes in the DNA repair machinery. These same genes have also been explored for therapeutic applicability such as DNA-PKs, ATM, PNKP and DNA ligase IV [21,25–28]. However, the ways that tumor cells deal with DNA damage is complex, and unexpected genes and pathways may be uncovered that impact radiation response, leaving cells with escape routes that might also be tumor-type specific. In addition, radiation has more impact on cells than DNA damage alone.

Consequently, radiotherapy response appears difficult to predict and many targeted radiosensitizers have not found their way to the clinic yet. Identification of novel genes or pathways contributing to radiosensitivity may reveal new leads for radiosensitizing therapies. The emergence of the CRISPR-Cas9 genome editing technique has enabled the possibility to study gene-therapy interactions unbiased and in great detail [29–31]. While previous techniques for gene function editing (either knockout or knockdown) had restrictions in scalability, durations, and specificity in effects, CRISPR-Cas9 screens have been shown to overcome these limitations and allow reliable genome-wide functional genomic interrogation. The aim of this study was to identify genes that can sensitize human HPV-negative HNSCC cells for treatment with radiotherapy by a genome-wide functional genetic approach.

## Materials and methods

### Cell lines

The UM-SCC-11B (RRID:CVCL_7716) cell line was a kind gift of Thomas Carey (University of Michigan, USA), and its identity was confirmed by STR profiling [32]. Cells were cultured as previously described [33]. UM-SCC-11B has been characterized as a typical HPV-negative HNSCC cell line with a heterogeneous CNV pattern, complete loss of *CDKN2A* and mutations in squamous cell carcinoma related cancer driver genes *TP53, NOTCH1, KMT2D and NSD1* [34]. Furthermore, UM-SCC-11B is known to be relatively radiation resistant [33].

Stable Cas9-expressing UM-SCC-11B cells (UM-SCC-11B Cas9) were established by transduction with a lentiviral vector with a blasticidin-resistance and Cas9 cassette (Addgene; #73310; RRID: Addgene_73310). Conditions for screening with lentiviral gRNAs were optimized, such as plating density, radiation dose and concentrations of polybrene and puromycin used for transduction and selection.

### Genome-wide CRISPR-Cas9 knockout library screen

We used the Toronto Knockout library version 1 (TKOv1) (Addgene; #1000000069) [29]; this pooled library consists of specific single guide RNAs (sgRNAs) encoding constructs in lentiviral vectors for gene knockout in the human genome. The library contains 91,320 sgRNAs, targeting a total of 17,661 genes (with mostly six sgRNAs targeting one gene), while several control guides are included targeting LacZ, EGFP, luciferase, and random loci on chromosome 10. The UM-SCC-11B Cas9 cell line was transduced with the TKOv1 library overnight at a low multiplicity of infection (MOI) (~0.3) to ensure a large proportion of cells with a single sgRNA integration, while minimizing cells with multiple integrations. Successfully transduced cells were selected with 1 µg/ml puromycin for 48 hours. Transduction and selection were controlled by puromycin selection without virus transduction (cells had to die) and by virus transduction without puromycin selection (cells had to stay vital).

At $T_0$ (after 48 hours puromycin selection), the transduced cell pool was split in two separate pools (treated and control) and seeded at optimal density in One-Well plates (Greiner Bio-One, Cat. No. 670180) to allow clonal expansion. A large number of cells was plated to provide a ~200-fold representation of the 90k library, while also anticipating a radiation effect (treated condition $2.2 \times 10^7$ cells and control condition $1.8 \times 10^7$ cells plated). Plates were incubated overnight, and the relevant plates were treated by γ-radiation with a dose of 2 Gray (Gy) using a $^{60}$Co source (Gammacell 220; MDS Nordion, Ontario, Canada), at room temperature. At days 4 ($T_4$) and 7 ($T_7$), the cells were trypsinized and replated at the baseline density, and a pellet of $1.8 \times 10^7$ cells representing ~200 fold the 90k library was collected for sequencing. At day 11 ($T_{11}$) all cells were harvested and stored in pellets of $1.8 \times 10^7$ cells. The screen was performed in two separate experiments.

### DNA isolation and sequencing of sgRNAs

Genomic DNA (gDNA) was extracted using the QIAamp Blood Maxi kit (Qiagen, Cat. No. 51192) according to the manufacturer's protocol. The sequencing libraries were prepared by performing a two-step PCR. First, 2 µg gDNA was amplified for each sample using the KAPA HiFi HotStart ReadyMix (Roche; Cat. No. 07958935001) in 40 parallel PCR reactions of 50µl followed by a second PCR reaction, adding the Illumina adapters and Illumina i5 and i7 indices. The PCR product was cleaned using the QIAquick PCR Purification Kit (Qiagen, Cat. No. 28104). The used PCR primers are listed in the S1 Table in S1 File. Resulting libraries were sequenced on a HiSeq4000 and reads were mapped to the TKOv1 library sequences without allowing mismatches.

### Screen analysis

The raw demultiplexed FASTQ files were uploaded in CRISPRAnalyzer (http://crispr-analyzer.dkfz.de/) [35]. Read counts obtained by CRISPRAnalyzer were used for further downstream analysis. We used the DrugZ pipeline to identify treatment-gene interactions by calculation of a Z-score per gene [31]. Both replicates were analyzed separately and a gene-irradiation interaction was assumed when the normZ score was at least below 0. Genes with a p-value <0.05 on both replicates were expected to have a significant reproducible effect, and therefore assigned as hit. Alternatively we used a Poisson distribution to estimate the confidence interval of depletion/enrichment rate of the read counts in the treated sample versus the control (S1 Methods in S1 File). The two hit lists were merged. Functional enrichment analysis was performed with ShinyGO 0.77, an online gene-set enrichment tool. [36]

### Establishment and functional analysis of knockout cell lines

The UM-SCC-11B[Cas9] cell line was used to produce knockout clones for validation of hits of the screen, *PRKDC* knockout cell clones were established as a positive control for the radiosensitizing effect [37]. In short, UM-SCC-11B[Cas9] cells were transfected with a CRISPR RNA (crRNA) of the different targets (See S2 Table in S1 File for specific sequences) and trans-activating crRNA (Dharmacon, U-002000–05) using Dharmafect 1 transfection reagent (Dharmacon, T-2001). Gene mutation status was confirmed by Sanger sequencing with BigDye™ Direct Cycle Sequencing Kit (Applied Biosystems, Cat. No.4458687) according to the protocol of the manufacturer. For details on the detailed establishment of the knockouts, clone selection by limiting dilution, western blot and functional analysis by clonogenic and etoposide assay, we refer to the supporting information.

### Staining and immunofluorescence microscopy of mitotic spindles

Spindle staining and microscopic glass slide preparation was performed as described previously [38] and immunofluorescence microscopy was performed using a Leica DM6000. Analysis and pole-to-pole measurements were performed using LAS Application Suite X software (version 3.4.2, Leica).

### Chromosome break analysis

Cells were plated in T-75 culture flask (Greiner Bio-One, Cat. No. 658175) at optimal density and cultured as mentioned above. When a confluency of 80% was reached, cells were either irradiated with 2 Gy or remained untreated. Cells were harvested, resuspended in 0.075 M KCl for 20 min, and fixed in methanol/acetic acid (3:1). Cells were washed in fixative three times, dropped onto glass slides, and stained with 5% Giemsa (Merck). Breaks were counted in 50 metaphases per condition on two different coded slides.

### Nocodazole assay

The Cas9-expressing UM-SCC-11B subclone and the *CLASP1* knock-out clone were seeded in T-25 flasks at $5\times10^5$ cells per flask, at day 4 cells were irradiated by γ-radiation with a dose of 2 Gy on a $^{60}$Co source (Gammacell 220; MDS

Nordion, Ontario, Canada) or remained untreated. Both conditions were either combined with or without 100nM nocodazole, after 16 hours cells were harvested for cell cycle analysis. Cells were pulse-labeled with 10µM 5'-ethynyl-2'-deoxyuridine (EdU) for ten minutes. After cell dissociation with trypsin and washing with PBS, cells were sequentially fixed in 2% para-formaldehyde, and ice-cold 70% ethanol and left overnight at −20°C. EdU-incorporation was detected using the Cu-coupled click reaction with the fluorophore Picolyl-azide 5/6-FAM (Jena Bioscience, Cat. No. CLK-1180). DNA content was measured using DAPI. Cells were analyzed on a BD LSR II Fortessa™ (BD Biosciences, Vianen, The Netherlands) using BD FACSDiva™ software (V8.0.1.1, BD Biosciences).

### DNA fiber assay

DNA fiber assay was performed as described in Van Schie et al. [39]

## Results

### Identification of genes in radiation response

To identify genes critical in the resistance to ionizing radiation in HNSCC, we performed a genome-wide CRISPR-Cas9 knockout screen in a cell line that is relatively radiation resistant. The workflow for the screen is depicted in Fig 1. Stable Cas9-expressing UM-SCC-11B cells (UM-SCC-11B$^{Cas9}$) were established by lentiviral transduction of a vector encoding a human-optimized Cas9 and a blasticidin-resistance cassette (Addgene plasmid #73310). After initial blasticidin selection, Cas9 expression was repeatedly tested by western blotting in consecutive passaged batches (S1A and S1B Figs in S1 File). Functionality of Cas9 was analyzed by transfecting UM-SCC-11B$^{Cas9}$ with a gRNA of a known essential target gene for HNSCC (PLK1). Cell viability of UM-SCC-11B$^{Cas9}$ after knockout of PLK1 using a gRNA showed a comparable effect to conventional siRNA knockdown of PLK1 in parental UM-SCC-11B cells (S1C Fig in S1 File). The radiation response (F = 0.29, p = 0.75; S1D Fig in S1 File) and growth rate of the transduced cell line was not compromised by introduction of the Cas9 vector.

We hypothesized that the optimal radiation dose for a screen is the dose for which the parental UM-SCC-11B shows minor response when compared to an isogenic control with a knockout of a well-known gene of radiation response PRKDC [37]. We therefore created a PRKDC knockout cell line (UM-SCC-11B $^{PRKDCko}$) by transfecting the UM-SCC-11B$^{Cas9}$ cell line with a PRKDC gRNA, performing a limiting dilution into single cell clones and verifying knockout clones by Sanger sequencing. A clonogenic assay showed an optimal window at 2 Gy y-radiation, with a relative surviving fraction of 0.69 (SD 0.04) for the parental cell line compared to a relative survival fraction of 0.07 (SD 0.04) for the PRKDC mutant cell line (S1D Fig in S1 File).

Subsequently, the TKO version 1 pooled library [29] was used to perform the genome-wide knockout screen. Lentiviral transduction was performed at a low MOI of 0.3 infectious units per cell to reduce the risk that cells receive multiple gRNAs. After puromycin selection, part of the mutant cell pool was pelleted to serve as pretreatment control ($T_0$). The remainder was split into two groups. One group was irradiated at a single dose of 2 Gy, while the other group constituted the control arm to identify the essential genes independent of radiation. Treatment was performed at day 1 after plating. Cells in both arms were passaged at day 4 and day 7, and collected at day 11 (Fig 1). This experimental setup, including pooled library virus transduction, was performed in duplicate.

Guide representation in the experimental arms was determined by amplifying the guide constructs from genomic DNA and enumerating all reads mapping to guides after next-generation sequencing of the amplification product. The median number of reads per sample and coverage per guide was $3.7 \times 10^7$ (Range: $3.6 \times 10^7 – 4.3 \times 10^7$) and 404 (390–474), respectively. For all samples the number of gRNAs with zero read counts was below 1%. There was high concordance between the normalized read counts of the mutant cell pools of both replicates (r = 0.85) (S2 Fig in S1 File). Screen performance was additionally assessed by analysis of previous described essential and non-essential genes [29]. As could be

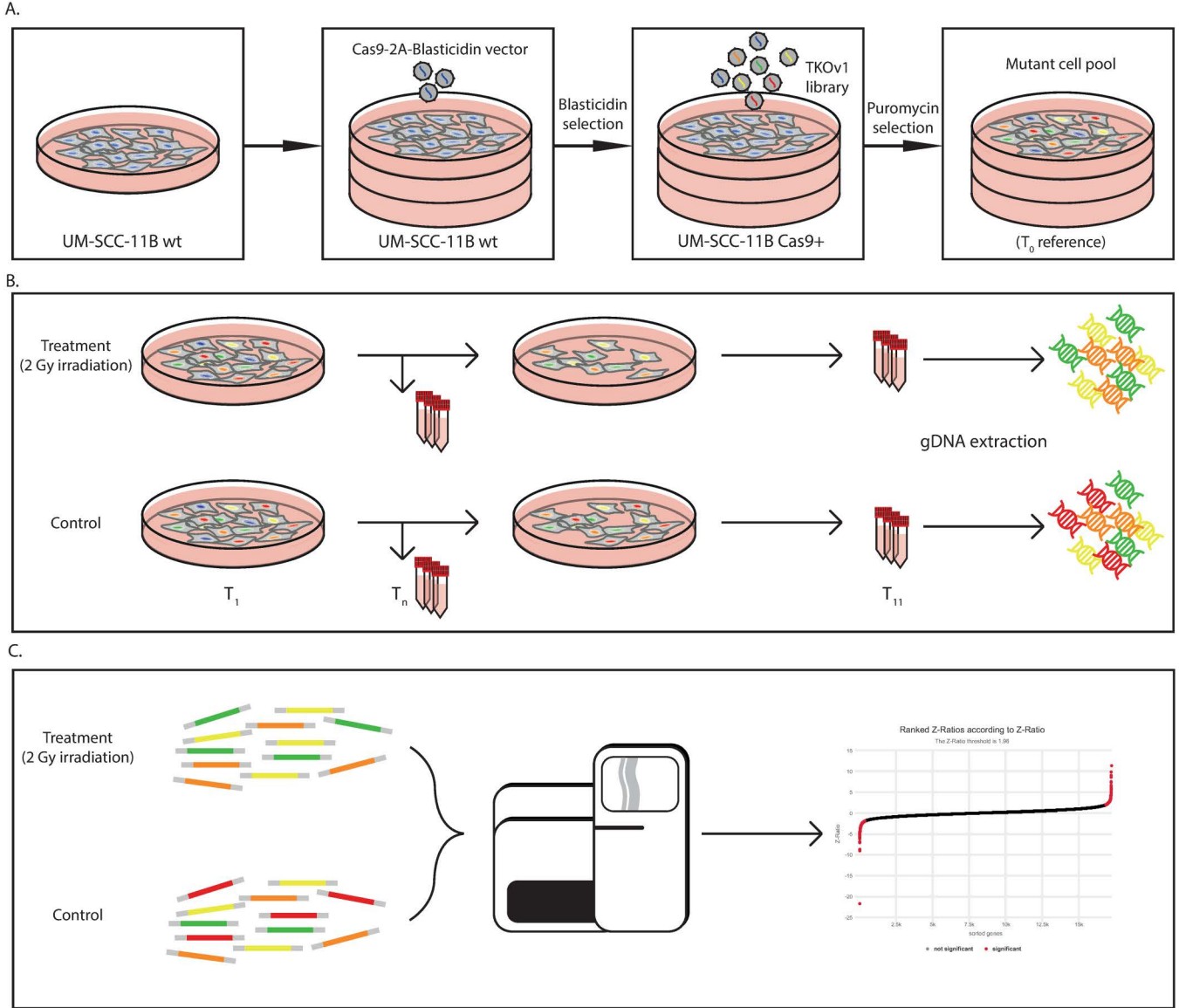

**Fig 1. Workflow of the CRISPR screening strategy.** The screen was performed in duplicate. **A)** A stable Cas9 expressing cell line was generated by transduction and blasticidin selection of HNSCC cell line UM-SCC-11B. The modified cell line was transduced with the TKOv1 knockout library with a MOI of ~0.3 and selected by puromycin to form a heterogeneous knockout pool ($T_0$). **B)** The puromycin selected cell pool was split in two, and after overnight incubation one pool was treated with 2 Gy γ-radiation, while the other pool was taken along as an untreated control ($T_1$). At day 4 ($T_4$) and day 7 ($T_7$) after plating the cells, the cells were split and replated. At $T_{11}$ cells were collected and genomic DNA was extracted for further analysis. **C)** sgRNA fragments were amplified by PCR and relative abundance of sgRNAs was determined by sequencing. Read-outs were analyzed by DrugZ and calculation of a normalized Z- score.

expected, the number of gRNAs targeting essential genes declined over time since knockout of these genes hampers the survival of the cells, causing a depletion (S3 Fig in S1 File).

We applied the DrugZ pipeline on both replicates separately for differential analyses of genes essential for survival after exposure to irradiation. Using a cut-off of $p < 0.05$, we identified in both replicates respectively 899 and 945 genes impacting radiation response. In total, 76 genes overlapped between both replicates (Fig 2A). As an alternative methodology,

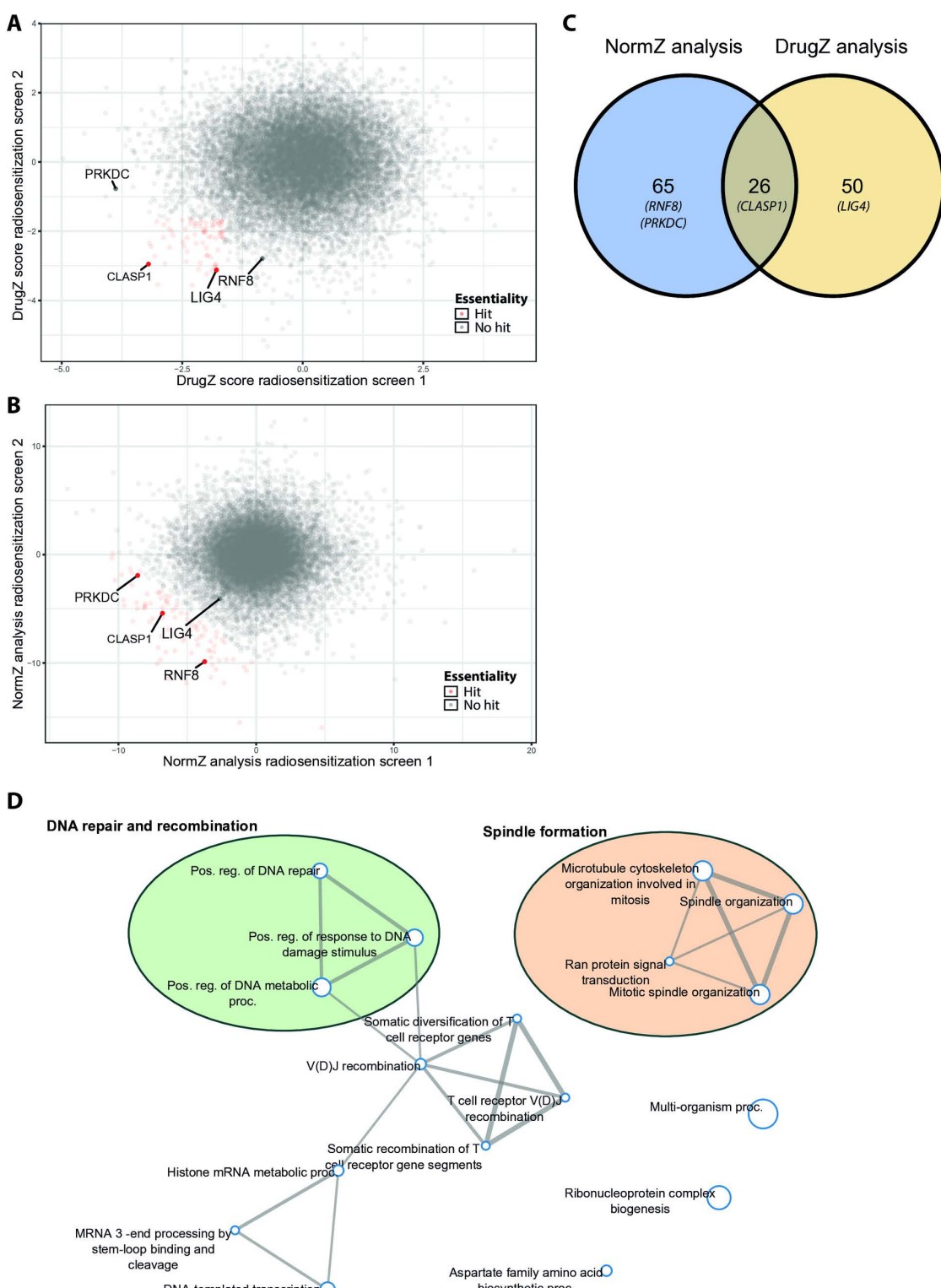

**Fig 2.** A) Normalized Z-scores calculated by DrugZ analysis of both replicates, red coloured points have a Z score ≤ 0 and a p-value ≤ 0.05 in both replicates. The radiosensitizing control (*PRKDC*) and genes selected for validation (*CLASP1, LIG4 and RNF8*) are highlighted. B) Normalized Z-score calculated for both replicates combined, red highlighted points have a combined NormZ score ≤ −10. C) Venn diagram showing the overlap in hits of both analysis pipelines (combined DrugZ and NormZ analysis. Genes selected for further validation are depicted in the diagram. D) Network plot showing

the relationship between enriched pathways, pathways (nodes) are connected if they share ≥20% genes. The FDR for enrichment was set at p<0.05. Darker nodes are more significant enriched pathways, and the size of the nodes represent the size of the gene set. This network plot was obtained by ShinyGO [36].

we calculated Z-scores and selected hits with a combined Z-score < −10 (S1 Methods in S1 File). By this analysis 91 genes were identified (Fig 2B), of which 26 overlapped with the DrugZ-identified hits (Fig 2C, S4 File). Next we merged the hit lists for further analyses. Functional enrichment analysis [36] on the 141 merged genes (91 + 76–26) showed, as expected, an enrichment (FDR<0.05) of genes involved in DNA repair (e.g., *ATM, LIG4, PRKDC, RNF8, FAM168A* and *DHX9)*(Fig 2D). *ATM* is a known gene active in radiation-induced DNA damage repair, but *LIG4* and *RNF8* are less well known genes in radiation response in HNSCC, although their role in DNA repair has been well studied in other tumor models [40–46]. Some of the genes impacted survival of the cells directly (depleted in T = 0 vs T = 11 in the parental cell lines).

We first tested *LIG4 and RNF8* for hit validation of the screens, while UM-SCC-11B $^{PRKDCko}$ was included as a positive control for the radiosensitizing effect. We generated knockout clones by transfection with respective sgRNAs and limiting dilution, and confirmed on-target cleavage of Cas9 by Sanger sequencing of single clones and protein analysis by western blotting (Fig 3A). Next we performed clonogenic assays to assess the effect of ionizing radiation on the knockout cell lines and compared it to our parental cell line. A linear-quadratic cell survival curve was fitted to the measured data, using the maximum likelihood estimation method [47]. The *RNF8* knockout clone showed high sensitivity to radiation, resulting in minor surviving fractions at 2 Gy (SF2) 0.11 ± 0.04 (*RNF8*), comparable with *PRKDC* (SF2: 0.07 ± 0.04) versus an SF2 of 0.69 ± 0.04 of the parental cell line. The cell survival curves were significantly different from the parental cell line (*PRKDC*: F = 66, p < 0.01 and *RNF8*: F = 164, p < 0.01; Fig 3B), confirming the role of these genes in radiation response in HNSCC. These data further verified the results of the genome-wide radiation screen. Unfortunately, the *LIG4* knockout cell line, although surviving the selection by limiting dilution, proliferated insufficiently to allow a clonogenic assay, and was therefore not further analyzed.

## Loss of *CLASP1* causes sensitivity to ionizing radiation

Besides DNA repair genes, genes involved in the formation of the microtubule network and the mitotic spindle were significantly more represented in our hits, such as *CLASP1, DCTN2, GPSM2, KIF23, KPNB1, CENPI, XPO1* and *STIL* (Fig 2D).

The top hit among these unexpected genes identified in our genetic screen was *CLASP1* (cytoplasmic associated protein 1). CLASPs have been identified on basis of their interaction with CLIPs involved in the plus-end dynamics of microtubule formation, and are as such involved in spindle formation but also focal adhesion [48]. Little is known about the interaction of *CLASP1* and ionizing radiation-induced cell death. To study the effects of *CLASP1* on the sensitivity to radiotherapy in more detail, we produced clonal *CLASP1* knockout cell lines from UM-SCC-11B$^{Cas9}$ using *CLASP1* gRNAs. After limiting dilution and clonal outgrowth, clones were sequenced and target modification was confirmed using the ICE v2 CRISPR Analysis Tool [49] (S4 Fig in S1 File). We produced three independent knockout cell lines of *CLASP1* confirmed by Sanger sequencing (S4 Fig in S1 File), protein expression analysis showed complete depletion of CLASP1 (Fig 3A).

The radiosensitizing effect of *CLASP1* was assessed by clonogenic assay, the *CLASP1* knockout cell lines showed reduced clonogenic capacities with a mean SF2 ranging from 0.36 (± 0.02) to 0.50 (± 0.13) compared to 0.69 (± 0.04) of the isogenic control (Fig 3B, p value <0.01). These results support the radiosensitizing effect of *CLASP1* knockout. Additionally, we performed a dose-response evaluation of our *CLASP1* clones and etoposide treatment. Etoposide is known to inhibit topoisomerase II causing DSBs alike ionizing radiation. IC50 determination assays were always measured in triplicate and performed in three separate experiments. The *CLASP1* knockout cell lines have an increased sensitivity to etoposide with IC50 values ranging from 0.295 μM to 0.414 μM compared to the isogenic control (0.866 μM).(S5 Fig in S1 File) The enhanced sensitivity was significant for *CLASP1* knockout clone 12 (p = 0.026) and *CLASP1* knockout clone 17 (p = 0.035). *CLASP1* knockout clone 18 was also more sensitive but not significantly different from the control (p = 0.067).

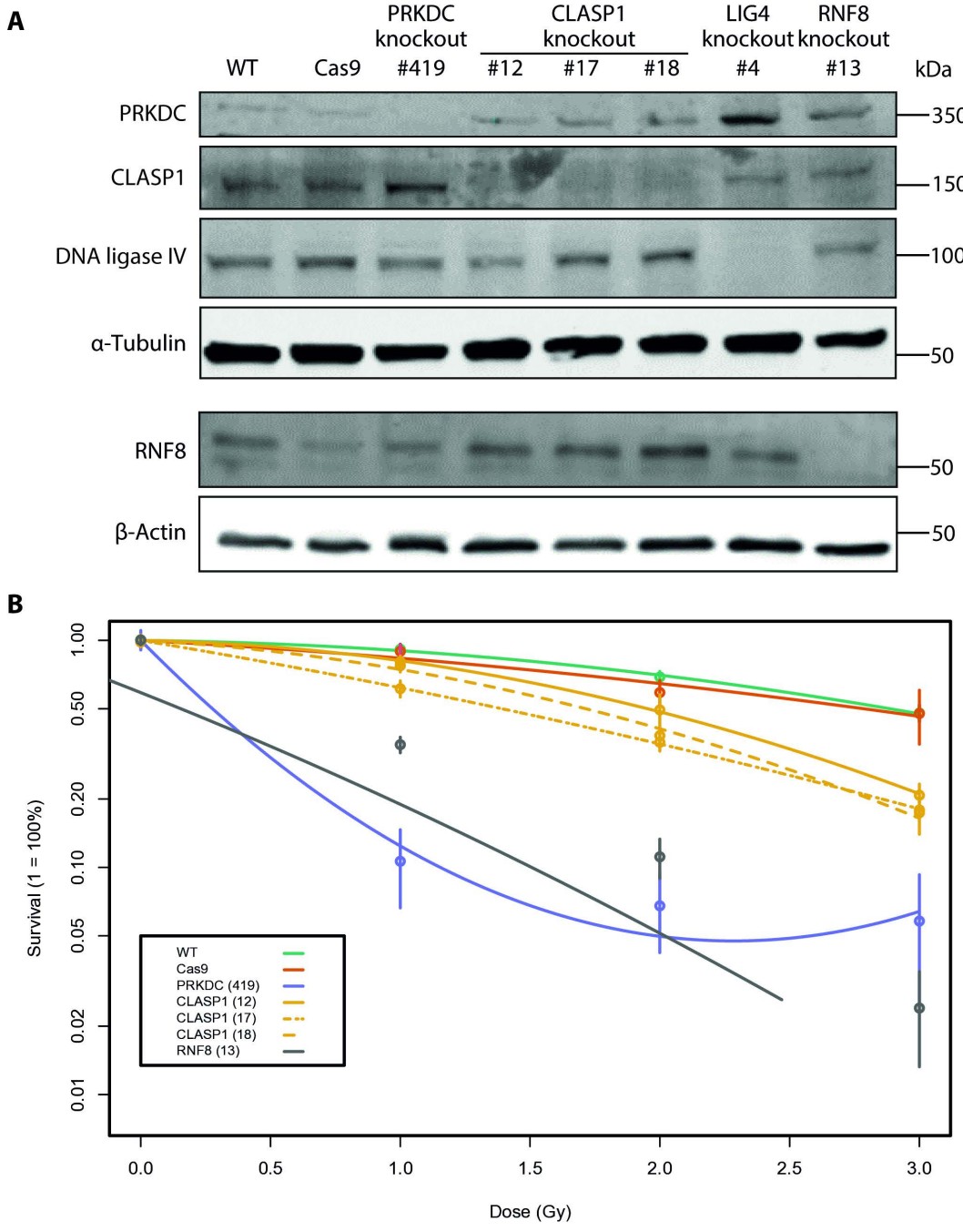

**Fig 3.** A) UM-SCC-11B<sup>Cas9</sup> cell line was used to generate single knockout clones by transfection with sgRNAs (*PRKDC, CLASP1, LIG4* and *RNF8*) followed by limiting dilution and clone selection, and confirmed on-target cleavage by Sanger sequencing of single clones. Western blot analysis confirmed functional knockout of genes. B) Clonogenic assays were performed to assess cell survival after ionizing radiation on the knockout cell lines, a linear-quadratic cell survival curve was fitted to the measured data.

Next we analyzed the role of CLASP1 mechanistically in the context of radiation. CLASP1 is best known as a microtubule (MT) stabilizing protein that acts at the plus ends of MT and serves as a rescue factor for MT catastrophe (49,50). We analyzed spindle formation and measured the pole-to-pole distance upon CLASP1 knockout using fluorescence microscopy. The mean pole-to-pole distance for the untreated parental cell line and the Cas9 cell line was 8.03±0.73 μm and 8.29±0.27 μm, a difference that was not significant (p=0.75). For CLASP1 (18) knockout cell line the pole-to-pole distance was significantly reduced to 5.89±0.13 (p<0.01) μm, respectively (Fig 4A), indicating that loss of CLASP1 impacts mitotic spindle formation as expected. However, this was not changed by radiation, suggesting that this activity of CLASP1 does not explain the observed increased radiation sensitivity.

Many radiosensitizing agents enhance the radiation effect by interference with replication stress and DNA repair-mechanisms of the cells, and impaired repair of radiation induced DNA damage leads to genomic instability and subsequent cell cycle arrest [25,50]. Inhibition of PRKDC, LIG4 and RNF8 are known to increase DNA damage and hamper cell cycle progression [37,42,51]. We therefore performed a chromosomal breakage assay with *PRKDC* knockout cell line as control. This chromosomal breakage analysis shows that spontaneous breaks occur sporadically and often only in low frequencies per cell, in both parental as in knockout cell lines. (Fig 4B) After irradiation, breaks occur more frequently as expected in 20% of mitotic cells of the parental cell line. In *PRKDC* knockout cell line almost all (96%) cells harbor chromosomal breaks after irradiation, while 82% of the cells has more than 10 breaks per cell. This could be explained by the compromised DNA repair. Remarkably, in the *CLASP1* knockout cell line we observed the same phenomenon, the majority of mitotic cells (60–84%) harbor chromosomal breaks of which 60–79% more than 10 breaks per cell after irradiation. That cells with impaired DNA repair showed increased chromosomal breaks is easily explained, but the knockout of *CLASP1* showing the same phenomenon was highly unexpected. To study changes in cell cycle distribution, we performed a BrdU assay with and without nocodazole, and with and without irradiation. Nocodazole is an anti-neoplastic drug that acts by depolymerizing the mitotic spindle, thereby arresting the cell cycle in the M-phase [52]. The assay revealed the reduced progression of *CLASP1* knockout cell line through S-phase particularly after nocodazole, and further increased by irradiation. In the parental cell line, 35% and 48% of the population remained in S-phase after 0 Gy and 2 Gy, respectively. In the *CLASP1* knockout cell line the progression though S-phase is decreased, with a remaining population of 63% and 72% in S-phase after 0 Gy and 2 Gy respectively, suggesting an early/mid S-phase delay or arrest. (Fig 5) These data strongly point for a role of *CLASP1* in S-phase progression. We therefore decided to perform a DNA-fiber assay to determine replication fork speed (Fig 6).

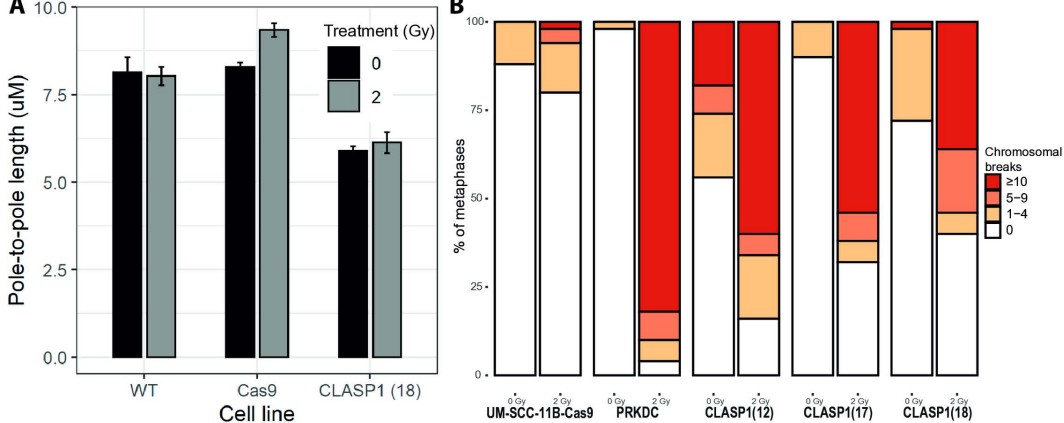

**Fig 4.** A) Pole-to-pole measurements of the mitotic spindles was carried out on the parental UM-SCC-11B cell line, the Cas9 transduced cell line and the *CLASP1* knockout cell line (UM-SCC-11B^CLASP1 (18)). B) Chromosomal breakage assay on the UM-SCC-11B^Cas9 cell line, the *PRKDC* knockout cell line the *CLASP1* knockout cell lines, with and without irradiation. Breaks were counted in 50 metaphases per condition.

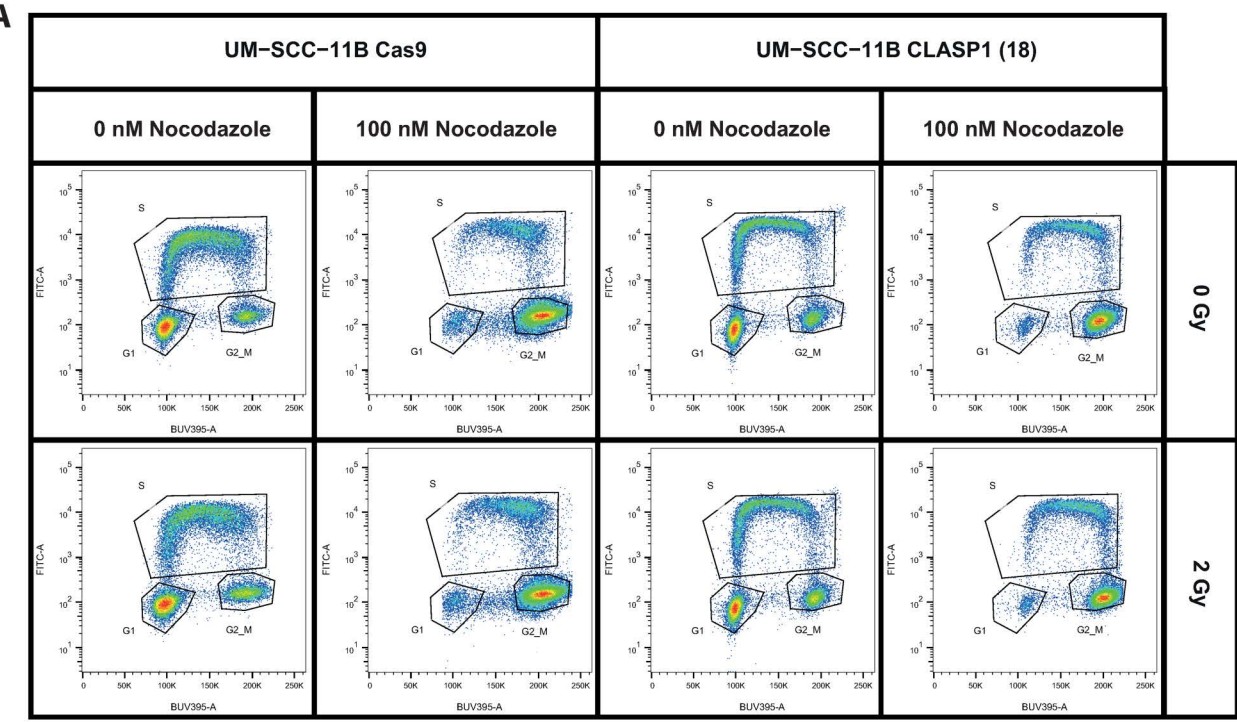

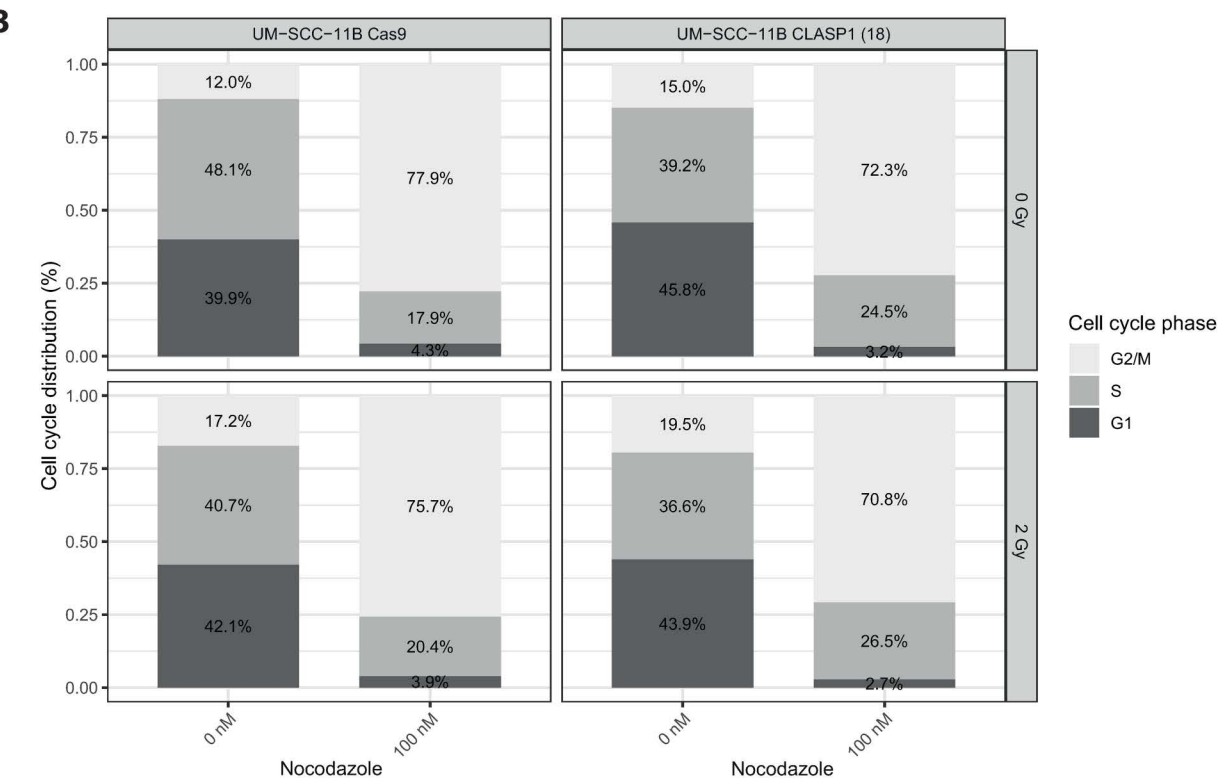

**Fig 5.** A) Cell cycle analysis of UM-SCC-11B[Cas9] and the *CLASP1* knockout cell line (UM-SCC-11B[CLASP1 (18)]), cell cycle analyses was performed with EDU and with and without treatment (2 Gy ionizing irradiation). Both conditions were either combined with or without 100nM nocodazole. B) The various cell populations of the cell cycle with the different conditions are depicted as bar graph. Particularly after nocodazole treatment a delay in S-phase is visible after *CLASP1* knockout.

After irradiation, fork speed decreased dramatically, and DNA fiber lengths could not be determined with confidence. Without irradiation, the effect between parental cell line and *CLASP1* knockout cell line is significant. Removal of *CLASP1* has major impact on replication speed, increased replication stress and causes chromosomal breaks. This is aggravated by irradiation. Whether this role of *CLASP1* in DNA replication still relates to microtubule network regulation or point to an independent novel direct activity of *CLASP1*, remains to be elucidated.

## Discussion

Whole genome CRISPR screens offer a unique opportunity for unbiased identification of genes essential for treatment response. We identified 141 genes essential for radiation response in HNSCC. We confirmed the sensitizing effect of the expected genes involved in DNA damage repair (*LIG4, RNF8 and PRKDC*) in ionizing radiation response. In addition, we identified a variety of genes active in mitotic spindle assembly or microtubule dynamics in broader sense, amongst which *CLASP1.* Our study on *CLASP1* revealed the role of this gene in response to radiation induced DNA damage.

Classically, genes associated with response of the cell to DNA damage play a key role in sensitivity to irradiation in HNSCC [21,25,26]. The exact nodes may differ per tissue type, but here we identified several including *RNF8* and *LIG4*.

RNF8 (Ring Finger Protein 8) is an important factor in the repair of DNA DSBs by initiating ubiquination of proteins at the damaged chromatin, and thereby starting recruitment of downstream factors (such as 53BP1, RAD51, and BRCA1) which results in the restriction of DNA end resection, thereby promoting NHEJ [40]. As the induction of DNA DSBs is considered the main effect of ionizing radiotherapy, several studies emphasize the importance of RNF8 in the response to radiotherapy [41–43]. Our data demonstrate the vital role of RNF8 in the response to ionizing radiation in HNSCC as well, suggesting that further studies on RNF8 targeting agents could lead to enhancement of radiotherapy.

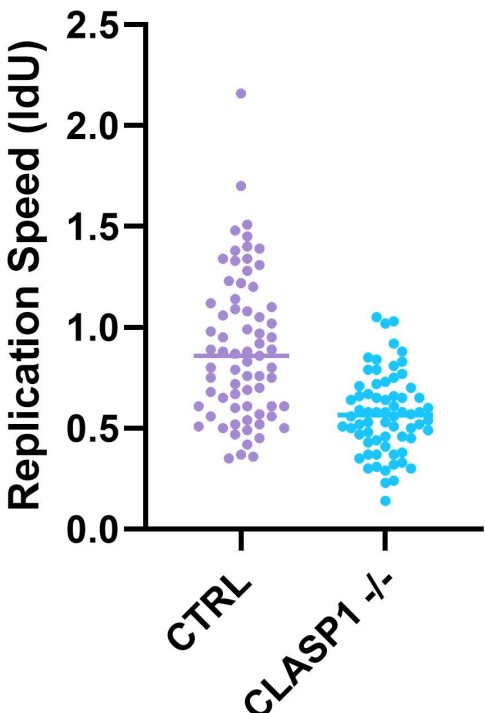

**Fig 6. DNA fiber assay of parental and *CLASP1* knockout cell line (UM-SCC-11B**[CLASP1 (18)]**).** Absence of CLASP1 has a negative effect on replication speed.

DNA ligase IV (encoded by *LIG4*) executes the final ligation step critical to DSB repair by NHEJ. Germline mutations in this gene result in the LIG4 syndrome, which is characterized by radiosensitivity, immunodeficiencies, developmental abnormalities and predisposition for cancer [51,53]. In the process of NHEJ, Ku70 and Ku80 proteins quickly join the blunt DNA ends to form a Ku-DNA complex. LIG4 acts in a complex with XRCC4, this complex binds to Ku-DNA complex, and without interference of any other factors ligates both DNA ends together [44]. Although we were not able to validate *LIG4* with a clonogenic assay as an essential gene for radiation response as its knockout already negatively impacted clonogenic growth, our screen results support the importance of *LIG4* in response to radiotherapy and suggests that targeting LIG4 or XRCC4 could have a radiosensitizing effect in HNSCC.

The identification of CLASP1 was remarkable. CLASP1 is best known as a MT stabilizing protein, acting at the plus ends of MT and serving as a rescue factor for MT catastrophe [54]. Additionally, during mitosis CLASP1 seems to be a key component in the interaction of the chromosome kinetochore with MTs [55,56], thereby contributing to satisfy the spindle assembly checkpoint (SAC) and ensuring the continuation of the mitosis [57]. CLASPs are also known for their role in non-centrosomal MT-formation at the trans-Golgi network during interphase, and thereby interfere with intracellular trafficking [58]. Several studies show the role of MT in the interphase in nuclear organization and genome stability [59]. The *CLASP1* gene is rarely affected in HNSCC, with reported mutation frequency of 0–1.5% (numbers obtained through cBioPortal) [60–63] and is amplified in ~1% of cases [63]. *CLASP1* is not differentially expressed in tumor versus mucosa in HNSCC [64]. Hence, the identification of *CLASP1* as radiation response mediating gene, is highly remarkable.

Knockout of *CLASP1* leads to significant reduction of pole-to-pole length due to inadequate formation of the MT network, but this effect is not influenced after exposing cells to irradiation. Strikingly, our cell cycle analysis and chromosome break analysis suggests that the radiosensitizing effect of *CLASP1* is mainly due to interference with the progression in S-phase. This would imply an unforeseen function for *CLASP1*, potentially in DNA replication. Whether this relates to replication speed or resolving stalled replication forks, remains to be determined. Poruchynsky et al. previously described the role of interphase MT in trafficking of DNA damage-repair proteins [65]. Although Efimov et al. showed a role for CLASPs in non-centrosomal MT-formation and trafficking of proteins in interphase [58], the specific mechanism of CLASP1 and the microtubule network in S-phase is unclear at present.

Our study demonstrates the power of CRISPR screens to identify unexpected mechanisms, but obviously meets limitations. In general, as mentioned by Dempster et al., choosing the right sgRNA library and the duration of the screen influences the efficacy of the screen [66]. Furthermore, CRISPR screens are thought to have a false-negative rate of around 20%, which can be mitigated by increasing the number of replicates. By extending the course of our experiment or increasing the number of replicates, we would in theory be able to pick up more radiotherapy enhancing genes with more moderate effects. In addition, our screen in cultured cells did not reveal the other typical factors in radiation response: hypoxia and EMT. Performing such screens in xenograft models might provide a more complete picture, although this also meets limits and is technically challenging. Nevertheless, we were able to find the well-known genes responsible for radiation effect (e.g., *PRKDC*, *ATM* and *LIG4*), and validated several promising hits.

In conclusion, we identified *CLASP1* as radiosensitizing target. Inhibition of CLASP1 impacts replication speed and S-phase progression, increases DNA damage leading to chromosomal breaks, aggravated by irradiation. Whether this relates to microtubule dynamics in general, or an independent function of CLASP1 in DNA replication, remains to be elucidated. In addition, whether this is a phenomenon typical for malignant HNSCC cells warrants further studies. The G1/S cell cycle checkpoint and the DNA damage response are inactivated in HNSCC cells, while these are intact in normal cells. These unexpected proteins and pathways offer potential as an alternative to conventional systemic agents (e.g., cisplatin), with potentially less toxicity due to selective disruption of spindle biology and progression from G2 to M-phase. To date there are no publicly available inhibitors of CLASP1. Our results warrant future endeavors aimed at developing such an inhibitor to pave the way for further pre-clinical validation.

## Supporting information

**S1 File. Supporting information on materials and methods, and supporting figures.**
(DOCX)

**S2 File. Western Blot raw data.**
(PDF)

**S3 File. CRISPR-Cas9 screen analysis raw data.**
(XLSX)

**S4 File. Hit list from the CRISPR-Cas9 screen.**
(XLSX)

## Acknowledgments

We would like to thank Jaap van den Berg for his support in performing the clonogenic assays and Idil Kirdök for the fiber assay analysis.

## Author contributions

**Conceptualization:** Reinout H. de Roest, Ruud H. Brakenhoff.

**Formal analysis:** Reinout H. de Roest, Arjen Brink, Jos B. Poell.

**Investigation:** Reinout H. de Roest, Marijke Buijze, Myrthe Veth, Klaas de Lint, Govind Pai, Martin A. Rooimans, Arjen Brink.

**Methodology:** Reinout H. de Roest, Marijke Buijze, Myrthe Veth, Klaas de Lint, Govind Pai, Martin A. Rooimans, Rob M.F. Wolthuis, Jos B. Poell, Ruud H. Brakenhoff.

**Resources:** Klaas de Lint.

**Supervision:** Ruud H. Brakenhoff.

**Writing – original draft:** Reinout H. de Roest.

**Writing – review & editing:** Klaas de Lint, Govind Pai, Rob M.F. Wolthuis, Arjen Brink, Jos B. Poell, Ruud H. Brakenhoff.

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
