## [Decision Letter · Decision Letter 0]

3 Jun 2025

Dear Dr. de Roest,

Thank you for submitting your manuscript to PLOS ONE. After careful consideration, we feel that it has merit but does not fully meet PLOS ONE’s publication criteria as it currently stands. As the reviewers pointed out, additional results and significant reorganization of the current results are needed before we can move forward. Therefore, we invite you to submit a revised version of the manuscript that addresses the points raised during the review process.

We look forward to receiving your revised manuscript.

Kind regards,

Zhiming Li, Ph.D.

Academic Editor

PLOS ONE

Journal Requirements:

“RHB

KWF-A6C7072 (Design project)

KWF Dutch Cancer Society

https://www.kwf.nl/en/dutchcancersociety

No”

4. Please include a caption for figure 3.

Reviewers' comments:

Reviewer's Responses to Questions

**Comments to the Author**

1. Is the manuscript technically sound, and do the data support the conclusions?

Reviewer #1: Yes

2. Has the statistical analysis been performed appropriately and rigorously?

Reviewer #1: Yes

3. Have the authors made all data underlying the findings in their manuscript fully available?

Reviewer #1: No

4. Is the manuscript presented in an intelligible fashion and written in standard English?

Reviewer #1: Yes

Reviewer #1: Reinout H. de Roest and colleagues did a great job conducting the genome-wide CRISPR screening to identify genes that sensitize HNSCC to radiation. The article is well written and clear to the readers. Experiments are mostly carefully planned and executed. However, some modifications or extra work is needed to make it suitable for publication.

1. FigS1A Cas9 blot is described to be repeatedly tested in consecutive passaged batches, but only one band/sample is shown in the figure which is not sufficient for such description. Also, for a proper blot, parental UM-SCC-11B cells should be used as reference to show Cas9 expression after Blast selection.

2. Figure number is a bit messy in the manuscript. Usually, the number should be consecutive in the order they are mentioned in the text (especially for the main figures). E.g. Fig 3B is mentioned earlier than Figure 2 in the text. Fig 4B is mentioned before Fig 4A.

3. It is great that the authors carefully calibrated the radiation dose through clonogenic assay and proper control (PRKDC mut cells), however the cell culture duration after radiation are not consistent in different experiments(clonogenic assay vs CRISPR screening). Notably, it is usually recommended to analyze cells from CRISPR screening after 10 doublings (not 10 days) to allow sensitive detection of genes with moderate defects, especially for negative selection screens (e.g. with radiation). UM-SCC-11B with about 31hr doubling time, would need 14 days in this case.

4. The coverage for the screening is a bit low (~200 representations) which may be compensated by the big library size. But two experimental replicates also restrict the reliability of the results(which is also discussed in the manuscript). It would be best if the authors could conduct a third biological replicate for more convincing results.

5. Fig 5A/5B should be labeled with 100nM Nocodazole instead of just 100nM, which is confusing. The paragraph describing Figure 5(line 310-332) needs to be re-written and Figure 5 should be reorganized (I don’t see any PRKDC related data here) for better understanding.

6. This is a nice research paper with well executed experiments. However, I have a general question for identifying CLASP1 as radiation sensitizing gene: regardless of the shortage of CLASP1 inhibitors, and not as significant sensitizing effect (compared to PRKDC and RNF8), how do you think the CLASP1 inhibition can change the normal cell sensitivity to radiation therapy? Also, would other CLASP protein/gene play a role in this process (e.g. making it less or more sensitive)?

**Do you want your identity to be public for this peer review?** For information about this choice, including consent withdrawal, please see our Privacy Policy

Reviewer #1: No

---

## [Author Response · Author response to Decision Letter 1]

3 Jul 2025

1. FigS1A Cas9 blot is described to be repeatedly tested in consecutive passaged batches, but only one band/sample is shown in the figure which is not sufficient for such description. Also, for a proper blot, parental UM-SCC-11B cells should be used as reference to show Cas9 expression after Blast selection.

We added the blot of the Cas9 expression of all consecutive batches of one of the replicates to the supplementary files, with the parental cell line as reference. The Cas9 expression after loading equal amounts of 45ug protein for all samples, remained stable over all consecutive passaged batches.

2. Figure number is a bit messy in the manuscript. Usually, the number should be consecutive in the order they are mentioned in the text (especially for the main figures). E.g. Fig 3B is mentioned earlier than Figure 2 in the text. Fig 4B is mentioned before Fig 4A.

The consecutive order of the Figures has been adapted.

3. It is great that the authors carefully calibrated the radiation dose through clonogenic assay and proper control (PRKDC mut cells), however the cell culture duration after radiation are not consistent in different experiments(clonogenic assay vs CRISPR screening). Notably, it is usually recommended to analyze cells from CRISPR screening after 10 doublings (not 10 days) to allow sensitive detection of genes with moderate defects, especially for negative selection screens (e.g. with radiation). UM-SCC-11B with about 31hr doubling time, would need 14 days in this case.

We thank the reviewer for the positive remarks. Of note, clonogenic assay requires a different approach compared to cell culture assays since in clonogenic assays we seed with limited dilution to obtain single growing colonies, which necessitates a longer assay compared to cell culture. The lag phase before cells start to divide is very long for single cells, and many squamous cell lines refuse to grow in clonogenic assays. Hence, such a set-up is not possible for a CRISPR screen.

We agree with the reviewer that a longer screening assay would result in potentially more sensitive detection of genes with moderate effect, however we were aiming for genes with larger effect and comparable to our positive control (PRKDC). We hypothesized that the length of our assay was sufficient to identify these genes. We added a statement on this in the Discussion.

4. The coverage for the screening is a bit low (~200 representations) which may be compensated by the big library size. But two experimental replicates also restrict the reliability of the results(which is also discussed in the manuscript). It would be best if the authors could conduct a third biological replicate for more convincing results.

As mentioned in our discussion, we agree with the reviewer that an additional replicate could lead to less false positive hits. However, irradiation is logistically much more difficult than applying drugs, and we therefore had to choose for a screen in duplicate. The robust analysis applied reduces false-positives.

5. Fig 5A/5B should be labeled with 100nM Nocodazole instead of just 100nM, which is confusing. The paragraph describing Figure 5(line 310-332) needs to be re-written and Figure 5 should be reorganized (I don’t see any PRKDC related data here) for better understanding.

We updated the labels in Figure 5, as suggested by the reviewer. The paragraph mentioned by the reviewer was referring to Figure 4 and not 5, a mistake that has been corrected in the revised manuscript.

6. This is a nice research paper with well executed experiments. However, I have a general question for identifying CLASP1 as radiation sensitizing gene: regardless of the shortage of CLASP1 inhibitors, and not as significant sensitizing effect (compared to PRKDC and RNF8), how do you think the CLASP1 inhibition can change the normal cell sensitivity to radiation therapy? Also, would other CLASP protein/gene play a role in this process (e.g. making it less or more sensitive)?

We thank the reviewer for the positive remarks and obviously agree: the finding of CLASP1 was highly remarkable. We could explain why the cells die by increased S-phase replication stress that lead to DNA damage, but to unravel this unexpected mechanistic role of CLASP1 and subsequent other proteins will require a tremendous amount of subsequent work, which might also include primary fibroblasts or keratinocytes as normal cells to assess the effect .

The CLASP family consists of CLASP1 and CLASP2, CLASP2 was not a hit in our screen and knock-out of CLASP2 did not give a strong radiosensitizing effect .The data were not mature enough to add it to the MS in our view and requires further experimentation.

---

## [Decision Letter · Decision Letter 1]

22 Jul 2025

An unexpected role of CLASP1 in radiation response and S-phase regulation of head and neck cancer cells

PONE-D-25-24444R1

Dear Dr. de Roest,

We’re pleased to inform you that your manuscript has been judged scientifically suitable for publication and will be formally accepted for publication once it meets all outstanding technical requirements.

Kind regards,

Zhiming Li, Ph.D.

Academic Editor

PLOS ONE

Additional Editor Comments (optional):

Reviewers' comments:

Reviewer's Responses to Questions

**Comments to the Author**

Reviewer #1: All comments have been addressed

2. Is the manuscript technically sound, and do the data support the conclusions?

Reviewer #1: Yes

3. Has the statistical analysis been performed appropriately and rigorously?

Reviewer #1: Yes

4. Have the authors made all data underlying the findings in their manuscript fully available?

Reviewer #1: Yes

5. Is the manuscript presented in an intelligible fashion and written in standard English?

Reviewer #1: Yes

Reviewer #1: Reinout H. de Roest and colleagues addressed most of my questions. For example, it is great that authors were able to provide the Cas9 blot to show the consistent Cas9 expression. However, it could have been better to provide the tubulin blot along with the multi-sample blot for reference (tubulin is loading control which is meaningless to show for just one sample). Furthermore, I understand that some experiment work may not be so mature to be shown in the paper (like the CLASP2 related tests), but it will be a good add-on to the discussion to show the accuracy of the screening. The last thing I want to point out is that Figure 5 is attached after Figure 6 in the current version of manuscript. Otherwise, I think the manuscript is suitable for publishment after some proofreading.

**Do you want your identity to be public for this peer review?** For information about this choice, including consent withdrawal, please see our Privacy Policy

Reviewer #1: No

---

## [Editor Report · Acceptance letter]

PONE-D-25-24444R1

PLOS ONE

Dear Dr. de Roest,

I'm pleased to inform you that your manuscript has been deemed suitable for publication in PLOS ONE. Congratulations! Your manuscript is now being handed over to our production team.

Kind regards,

on behalf of

Dr. Zhiming Li

Academic Editor

PLOS ONE